# Novel Exon-Skipping Therapeutic Approach for the DMD Gene Based on Asymptomatic Deletions of Exon 49

**DOI:** 10.3390/genes13071277

**Published:** 2022-07-19

**Authors:** Mario Abaji, Svetlana Gorokhova, Nathalie Da Silva, Tiffany Busa, Maude Grelet, Chantal Missirian, Sabine Sigaudy, Nicole Philip, France Leturcq, Nicolas Lévy, Martin Krahn, Marc Bartoli

**Affiliations:** 1Medical Genetics Department, Assistance Publique Hôpitaux de Marseille, La Timone Children’s Hospital, 13005 Marseille, France; mario.abaji@ap-hm.fr (M.A.); svetlana.gorokhova@univ-amu.fr (S.G.); tiffany.busa@ap-hm.fr (T.B.); chantal.missirian@ap-hm.fr (C.M.); sabine.sigaudy@ap-hm.fr (S.S.); nicole.philip@ap-hm.fr (N.P.); nicolas.levy@univ-amu.fr (N.L.); martin.krahn@univ-amu.fr (M.K.); 2MMG, INSERM, Aix Marseille University, 13385 Marseille, France; nathalie.dasilva@univ-amu.fr; 3Centre Hospitalier Inter-Communal Toulon-La Seyne, Medical Genetics Unit, Sainte Musse Hospital, 83100 Toulon, France; maude.grelet@ch-toulon.fr; 4Department of Medical Genetics, APHP Centre Université Paris Cité Cochin Hospital, 75014 Paris, France; france.leturcq@aphp.fr

**Keywords:** genetics, pathological mechanisms, DMD, deletion, exon skipping, AON, muscular dystrophy, myopathy, therapy

## Abstract

Exon skipping is a promising therapeutic approach. One important condition for this approach is that the exon-skipped form of the gene can at least partially perform the required function and lead to improvement of the phenotype. It is therefore critical to identify the exons that can be skipped without a significant deleterious effect on the protein function. Pathogenic variants in the DMD gene are responsible for Duchenne muscular dystrophy (DMD). We report for the first time a deletion of the in-frame exon 49 associated with a strikingly normal muscular phenotype. Based on this observation, and on previously known therapeutic approaches using exon skipping in DMD for other single exons, we aimed to extend the clinical use of exon skipping for patients carrying truncating mutations in exon 49. We first determined the precise genomic position of the exon 49 deletion in our patients. We then demonstrated the feasibility of skipping exon 49 using an in vitro AON (antisense oligonucleotide) approach in human myotubes carrying a truncating pathogenic variant as well as in healthy ones. This work is a proof of concept aiming to expand exon-skipping approaches for DMD exon 49.

## 1. Introduction

Duchenne muscular dystrophy (DMD) is an X-linked recessively inherited genetic disease caused by the absence of dystrophin due to loss-of-function variants in *DMD* gene. It is the most common muscular dystrophy in children with an incidence of 1 in 4000 male births [1] and represents a severe and progressive condition affecting boys who lose walking ability by the age of 12 years old (yo) due to muscle damage and develop cardiomyopathy and restrictive respiratory failure leading to severe disabilities and premature death [2]. A milder form, called Becker muscular dystrophy (BMD), develops when the causing mutations lead to the production of an incomplete protein, as compared to DMD where it is severely reduced or absent.

Dystrophin is a 427kDA cytoskeletal protein located mainly on the sarcolemma of striated muscles. It has a rod-like shape [3] linking the intracellular network to the membrane and it is essential for proper muscle function (Figure 1a). Multi-exon deletions in *DMD* are considered the most common mutational mechanism responsible for DMD and BMD phenotypes, followed by point mutations [4,5]. When a deletion interrupts the reading frame the truncated protein will be very weakly or non-expressed due to Nonsense Mediated Decay (NMD), resulting in a severe DMD phenotype. Conversely, in-frame deletions usually result in a shortened but still functional protein causing milder DMD or BMD phenotypes. The value of the mutational reading frame in predicting a phenotype can achieve up to 88.8% (Positive Predictive Value) for exonic deletions [4]. However, although the effect of some mutations can be predicted, the final phenotype can be influenced by several other factors [6]. In addition, the reading frame rule is not always accurate in predicting phenotype. For example, patients carrying in-frame deletions of exons 45 to 47 had a moderate form of BMD and others carrying in-frame deletions of exons 45 and 46 developed a severe form of DMD [7]. Other factors can affect the absence of pathogenicity in these patients as well as the presence of modifier genes. Genome-wide association studies (GWAS) performed on dystrophinopathies databases have identified modifiers associated with age of gait loss, grip strength and response to corticosteroids [8,9,10].

The standard treatment for dystrophinopathies relies on corticosteroids, rehabilitation and assisted ventilation. However, this does not prevent disease progression and has undesirable side effects. In recent years, promising gene therapy approaches have been developed, notably using exon skipping [11]. This approach aims to recreate an open reading frame and correct the effect of pathogenic mutations by using antisense oligonucleotide sequences (AON). These AON penetrate the nucleus of muscle cells and specifically mask essential sites for the splicing process at the pre-mRNA level, resulting in a targeted deletion in one or several exons on the mature mRNA. The purpose is to enable the synthesis of a “BMD-like” protein that will have an induced deletion but will still be functional [12]. 

After encouraging preclinical results obtained using animal models [13], several clinical trials were conducted worldwide which led to commercializations of different AON molecules targeting exons 51, 53 and 45 [14,15,16]. Nevertheless, clinical observations depending on the reading frame rule highlight the variability in protein function after exon-skipping treatment depending on targeted exons and on which part of the protein these exons encode.

This research work constitutes proof of the concept of using an exon-skipping approach in patients carrying pathogenic variants in exon 49 based on clinical observation.

## 2. Materials and Methods

### 2.1. Patients

DNA from patients carrying a benign exon 49 deletion was available and stored at the APHM (Assistance Publique-Hôpitaux de Marseille, Marseille, France) Biological Resource Centre (BRC, Marseille, France). DNA was extracted with the consent of patients or their legal representative prior to collection for research and diagnosis. In total, whole blood DNA from five patients was used. Myoblasts were obtained from a DMD patient carrying a pathogenic truncating variant in exon 49, DMD(NM_004006.3): c.7186_7187insT (p.Thr2396Ilefs*14).

### 2.2. Ethical Statement

All protocols performed complied with the ethics guidelines of the APHM. Written-informed consents from patients or their legal representatives were collected for their genetic analysis.

### 2.3. Polymerase Chain Reaction (PCR) and Sequencing

The fragment containing the breakpoint junction was amplified using Q5 High-Fidelity DNA Polymerase (M0491S, NewEngland Biolabs, Evry-Courcouronnes, France) with the following primers. 5′-AATTGCAGGTCCAAGGTTTTC-3′ for the intron 48 primer, and 5′GTTTGCCATTTGACTTGCCAG for the intron 49 primer. To sequence the fragment of interest, DNA was purified and then reacted with Big Dye buffer (Taq polymerase, dNTP and fluorescent ddNTP)(4336697, Thermo Fisher, Villebon-Sur-Yvette, France) and specific primers (see above), followed by sequencing on the ABI PRISM 3130 Genetic Analyzer.

### 2.4. Cell Culture

Healthy human myoblasts, (AB678C53Q) immortalized by the Institute of Myology myobank, and myoblasts isolated from a DMD patient in the Department of Biology and Molecular Genetics, Cochin Hospital which carried the frameshift variant in exon 49, were cultured. Myoblasts were cultured in T75 flasks with proliferation medium (1/5 Medium 199 and 4/5 Dulbecco’s Modified Eagle Medium-Glutamax (31965-021)) containing 20% fetal calf serum (A4736301 Thermo Fisher, Villebon-Sur-Yvette, France), 25 µg/mL fetuin, 5 ng/mL hEGF, 0.5 ng/mL dFGF, 5 µg/mL insulin and 0.2 µg/mL dexamethasone, at 37 °C with 5% CO_2_. Passages were performed weekly using trypsin and a maximum of 15 passages to avoid cell drift.

### 2.5. Differentiation into Myotubes

Differentiation into myotubes was performed using 6-well plates. Three drops containing myoblasts at a concentration of 1000 cells per µL were placed in each well coated with gelatine for adhesion. Differentiation into myotubes was performed using a differentiation medium containing Dulbecco’s Modified Eagle Medium-Glutamax (31966-021) (DMEM) with insulin and gentamicin. Intermediate differentiation medium, which has the same components as the differentiation medium in addition to 7% of fetal calf serum, was added on day 7 of differentiation.

### 2.6. Antisense Oligonucleotides (AON)

To transfect myotubes, AON were designed by choosing complementary sequences to essential regions for the inclusion of exon 49 in the messenger RNA during splicing. These sequences are mainly represented by ESE (exonic splicing enhancers) and the splice acceptor sites. The predicted locations of these sites were obtained from HSF Genomnis data (hsf.genomnis.com2021/04/12). AON were ordered from Eurogentec (Angers, France) with the following sequences: CAGUUUCCUGGGGAAAAGAAC for the AON (A) and CGGUUGUUUAGCUUGAACUG for the AON (B). Both sequences were 2’-O-methylated on all bases and were phosphorothioate-modified on all bases except the last. Chemical modifications have been added to increase stability and nuclease resistance [11].

### 2.7. Transfection of Myotubes with AON

Six days after plating, wells were washed twice with DPBS (Dulbecco’s phosphate-buffered saline) and AON were transfected using Oligofectamine transfection reagent (12252011, Thermo Fisher). This reagent forms stable complexes with AON, allowing efficient transfection in a highly specific yet non-toxic way. Transfection was performed following the manufacturer’s protocol using 2 µL of Oligofectamine and 10 µL of 20 µM AON in each well (total volume of 1 mL), for the conditions with AON A+B, we used 5µL for each AON.

### 2.8. RNA Extraction and Reverse Transcription Polymerase Chain Reaction (RT-PCR)

RNA extraction from transfected and non-transfected myotubes was performed using the phenol/chloroform method after washing with DPBS. RT-PCR was performed on the extracted RNA using high-capacity cDNA reverse transcription kit (4368814, Thermo Fisher). The exon-skipping efficiency was verified by PCR and sequencing using the following primers: the forward primer in exon 47 5′ TCCCATAAGCCCAGAAGAGC-3′ and the reverse primer in exon 51 T TGTGTCACCAGAGTAACAGTCT. All cDNA was sequenced using Genewize services. Quantification was performed using band densitometry using ImageJ software.

## 3. Results

### 3.1. Clinical Observations

In the medical genetics department of Marseille’s La Timone Hospital, an in-frame exon 49 deletion was incidentally found using CGH-array requested for various indications, in five individuals from four different families belonging to the same ethnic Romani origin: two males (hemizygous) and three females, one homozygous and two heterozygous. They consulted the clinical department of medical genetics for various reasons:

Patient **A**, a boy born to related parents presenting global developmental delay since the age of 5 months, who started walking at 4 yo and had difficulty with standing from a sitting position at 5 yo. Muscle biopsy showed a level of dystrophin on Western blot similar to control, and CPK measurement was normal. Patient A is now 11 yo without any walking difficulties. Patient A carries a hemizygous deletion of *DMD* exon 49, inherited from his healthy mother. No genetic diagnosis explaining the developmental delay has been made so far for this patient.

Patient **B**, a girl referred for left body hemihypertrophy and a suspected Beckwith–Wiedemann syndrome at the age of 18 months, which was confirmed by methylation analysis of the 11p15 region. Patient B carries a heterozygous deletion of *DMD* exon 49, inherited from her healthy mother.

Patient **C**, a boy carrying the deletion, had CPK levels at the upper limit of normal at the age of 2 yo, and walking was acquired at the age of 3 yo.

Patient (**E**), mother of patient (**D**), carries the deletion in a homozygous state and presents with normal neuromuscular examination, as well as CPK dosage. Her father (an obligate carrier and hemizygous) was not examined but lived until his seventies without any muscular symptoms.

Thus, none of the individuals carrying this deletion showed signs of BMD or DMD. Delay in gait acquisition in patients A and C can most likely be attributed to developmental delay.

### 3.2. Genomic Position of the Observed Deletion

The precise deletion size of *DMD* exon 49 is important to the understanding of underlying mechanisms leading to phenotypical variability. The deletion was initially identified by array CGH as an incidental finding (last deleted probes chrX:31849630-31849689 and chrX:31874999-31875058), and it was subsequently confirmed using MLPA (Multiplex Ligand-dependent Probe Amplification). To determine the precise genomic position of the deletion in *DMD,* we performed long-range PCR using a series of primer pairs designed in the *DMD* intron 48 and intron 49 sequences in proximity to the last deleted probes as determined by the array CGH analysis (Figure 1b). PCR results are shown in supplementary data (Figure 2) and demonstrate a fragment of 750 bp amplified using DNA from the five mentioned patients. After sequencing this fragment, the exact size and precise genomic position of the deletion were determined to be chrX:31845184-31878209 (hg19/GRCH37), which corresponds to an exact size of 33,026 bp.

### 3.3. In Vitro Application of Exon 49 Skipping in WT and Mutated Myotubes

Based on the above clinical observation, it is possible to consider an exon-skipping approach targeted on exon 49 of DMD, which is 102 bp long and encodes 34 amino acid residues, in patients carrying truncating mutations in this exon. This will theoretically allow the expression of a shorter, but functional, dystrophin and cancel the pathogenic effect of mutations. To demonstrate the feasibility of exon 49 skipping we performed Iin vitro experiments on wild-type (WT) healthy human muscle cells first, then in mutated myoblasts carrying the DMD(NM_004006.3):c.7186_7187insT (p.Thr2396Ilefs*14) pathogenic variant, after differentiation into myotubes. Since dystrophin is expressed late in the development of myotubes, working on this cellular model was important. Two AON were used and transfected into myotubes using Oligofectamine: (AON A) covers the canonical acceptor splice site (ag), and (AON B) hybridizes to important sequences for recruitment of the splicing machinery, the ESE sequences (Figure 3a).

AON B was used alone when experiments were done in mutated myotubes since it was sufficient to introduce exon 49 skipping on its own in WT myotubes. Twenty-four hours after transfection, RNA extraction and then RT-PCR using primers in exons 47 and 51 were performed. The results (shown in Figure 3b) demonstrated successful skipping of exon 49 using AON B in both wild-type and mutated myotubes. In lanes where cells were transfected with AON B, we noticed the presence of the expected 500 bp fragment, and a second band of 400 bp that represents the same expected fragment without exon 49 (102 nucleotides) as shown on chromatograms. The proportion of skipped band is 63.28% compared to the non-skipped (36.72%) in the mutated myotubes. All skipped band proportions are indicated in the Figure 3b legend).

## 4. Discussion

This research project constitutes a proof of concept for the use of exon skipping in patients carrying pathogenic variants in exon 49 and may lead to clinical trials in the future.

It is important to note that there is no single molecule to treat all DMD patients carrying different types of mutations, but that each type can eventually be targeted with different AON (antisense oligonucleotide sequences). Therefore it is important to identify exons to skip, and the choice of targeted exons is based on two criteria: the number of patients in whom exon skipping could be beneficial, as well as the evidence that deleted exons targeted by exon skipping allow synthesis of a functional dystrophin [17]. In our study, we describe cases of asymptomatic deletions of exon 49, suggesting that this exon can be targeted by therapeutic exon skipping. At least 17 pathogenic variants within this exon have been described (ClinVar, LOVD-DMD and UMD-DMD), with certain of these variants present in multiple patients, thus justifying the need for therapeutic exon 49 skipping. Moreover, the R19 spectrin-like repetition is the fourth most mutated domain in the protein, with 7.9% of pathogenic variants and small rearrangements occurring in it (after the R17 repetition (23.1%, ABD1 (13%) and R18 (8%)).

Several isolated deletions of exon 49 of *DMD* have already been reported in the literature associated with both DMD and BMD phenotypes [18,19,20,21,22,23] (Appendix A). Notablythese reports did not include any specific phenotypic information about disease severity. Moreover, the precise genomic breakpoints were identified only in one case (Marey et al.) [24]. We then characterized a different, smaller exon 49 deletion that is, intriguingly, asymptomatic in several individuals. Our results suggest that the pathogenicity of exon 49 deletions should be interpreted with caution. Indeed, a recent study reported an isolated deletion of exon 49 as an incidental finding in a prenatal setting in a male fetus, classifying this deletion as pathogenic [25]. However, it is possible that depending on the deletion size and breakpoints, the individuals could have a much milder phenotype or even be asymptomatic. Exon 49 encodes for a portion of the spectrin-like repeat number 19 (R19, encoded by exons 48–50). R19 is part of the second membrane lipid-binding domain of dystrophin (Figure 1a). Thirty-four amino acids (aa) are deleted in this 1684 aa domain [26], suggesting that most of this domain remains intact. We thus expect that the in-frame deletion of exon 49 would have a minimal effect on the dystrophin protein function. The more severe DMD phenotype in a few patients with exon 49 deletion could thus possibly be due to interference with other molecular mechanisms and the larger size of the deletion. One hypothesis could be that loss of intronic regions containing regulatory elements can play a role in expression, but this needs to be explored with further functional studies using other models. Furthermore, as mentioned above, genetic modifiers may also play a role in the phenotype. Lastly, reports of more severe phenotypes associated with exon 49 deletion may also be explained by incorrect ascertainment of the deleted exons due to low resolution analysis.

Our results show that exon 49 can be efficiently skipped in human muscle cells. However, additional studies are required to confirm that the produced dystrophin is functional and able to rescue the muscular phenotype. Using the precise breakpoint data obtained in this study, gene editing in myoblasts could be used to recreate the deletion at the genomic level to decipher the molecular mechanisms underlying the phenotypic variability associated with this deletion.

In this work, exon 49 deletion of *DMD* has been reported in patients without a pathogenic muscular phenotype. The precise genomic position of this deletion was then identified using long-range PCR. Next, feasibility of skipping exon 49 was demonstrated using AON transfected in healthy and mutated human myotubes. This can be considered as a proof of concept to continue the work towards functional studies using new cellular models, then towards the development of more robust therapeutic AON with the aim of treating patients carrying pathogenic variants in exon 49.

## Figures and Tables

**Figure 1 genes-13-01277-f001:**
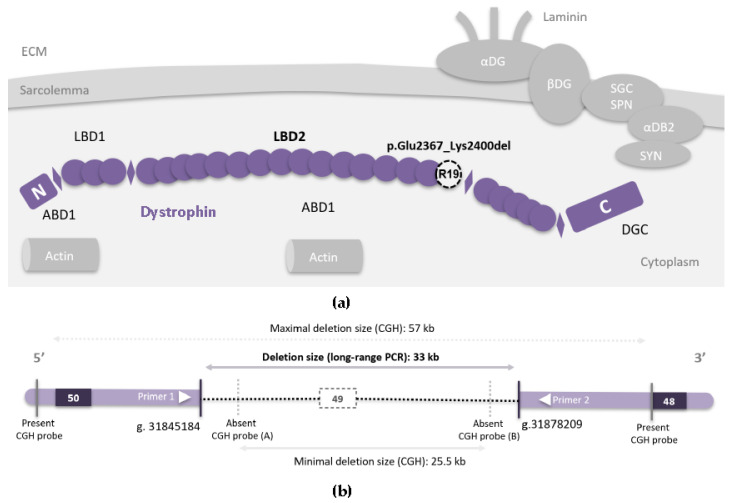
(**a**) Schematic representation of dystrophin (violet), a rod-shaped protein containing four main functional domains: an actin-binding amino-terminal domain (ABD1); a central rod domain composed of 24 spectrin-like repeats (R1–R24, represented by circles) interrupted by four proline-rich parts (H1–H4) which gives it more flexibility (represented by diamonds); a cysteine-rich domain; a carboxy-terminal domain. A second actin-binding domain (ABD2) extends from R11 to R17. Dystrophin has two membrane lipid-binding domains (LBD), the first one comprises the repeats R1 to R3 whereas the second one (LBD2) comprises repeats R4 to R19. This places dystrophin very near the sarcolemma with a large part of its central rod domain lying along the phospholipid membrane. R19 (LBD2), which is coded partially by exon 49, is represented by a dashed white circle. In the cellular context, dystrophin forms a complex with other proteins (DG: dystroglycans, SGC/SPN: sarcoglycan–sarcospan complex, DB: dystrobrevin, SYN: syntrophins). This complex plays an important role in signal transduction in addition to its mechano-protective role, which is indispensable for contractions and proper muscle function [3]. (DGC: dystrophin glycoproteins complex, ECM: extracellular matrix); (**b**) Schematic representation of exons 50, 49 and 48 in *DMD* (rectangles). The identified genomic deletion of exon 49 is represented by the dashed horizontal line. Genomic positions are mentioned using (hg19/GRCH37).

**Figure 2 genes-13-01277-f002:**
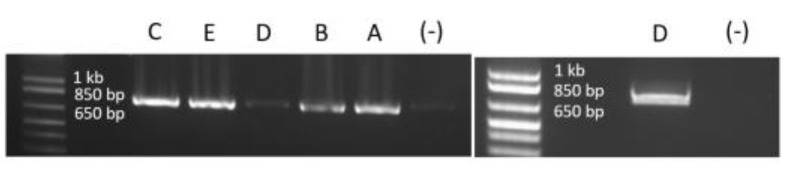
Long-range PCR results showing the amplified fragment containing the breakpoint junction of the exon 49 deletion, using Q5 High-Fidelity DNA Polymerase. We note that we have the same 750 bp fragments for the 5 individuals. All patients are identified by the same letter as in the text. (-) signs mark the negative control lanes (PCR amplification without DNA). For patient D (daughter of patient E), PCR had to be redone using a new source of DNA because of the degradation of the first source.

**Figure 3 genes-13-01277-f003:**
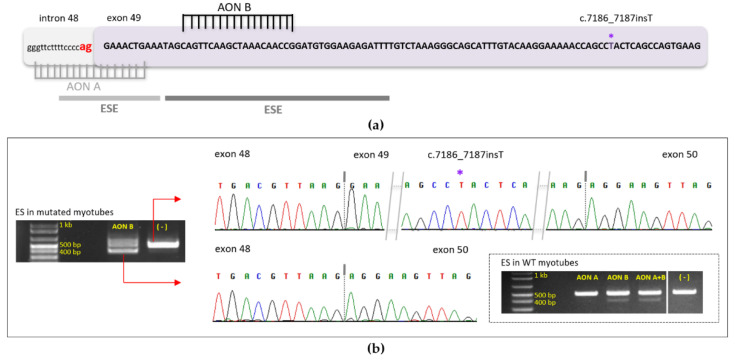
(**a**) Schematic representation of exon 49 DNA sequence in a DMD patient carrying a frameshift variant (asterisk). AON A and B, transfected in WT and mutated myotubes, are represented in grey and black respectively and hybridized to their complementary sequence. AON A masks the canonical (ag) splice site (red), AON B masks a region rich in ESE (exonic splicing enhancer), represented by horizontal lines (color reflects density of ESE presence); (**b**) Results of AON transfection into WT and mutated myotubes are presented. In the dashed rectangle: RT-PCR results after RNA extraction from transfected WT myotubes are shown on electrophoresis migration gel; primers located in exons 47 and 51 were used. In WT, myotubes bands without exon 49 represent 11.68% with AON A alone, 16.87% with AON B alone and 24.42% using AON A plus B. The second migration gel represents the results of transfected myotubes carrying the c.7186_7187insT pathogenic variant (RT-PCR). Exon skipping was done using AON B alone (first lane) and compared to the control (second lane). The band without exon 49 represents 63.28% of all amplified bands. The sequence of represented bands is shown with their respective chromatograms. (ES: Exon skipping, WT: Wild type).

## Data Availability

Data available on request.

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
