# Peer review of "Novel Exon-Skipping Therapeutic Approach for the DMD Gene Based on Asymptomatic Deletions of Exon 49"

_genes, 2022, doi:10.3390/genes13071277_

Round 1

Reviewer 1 Report

In this manuscript, the authors describe a number of patients of Roma origin who were incidentally found to have deletion of exon 49 of the DMD gene, but lack any neuromuscular manifestations. They subsequently designed an antisense oligonucleotide capable of inducing exon 49 skipping in vitro, and which could be a potential therapeutic agent for patients with truncating mutations in exon 49. I think that such reports of asymptomatic and minimally symptomatic DMD mutations are very valuable, both for counseling of patients and potentially for therapeutic developments. I do however have a few comments and suggestions to make, which I think would enhance the paper:

* Abstract: It would be helpful for readers to mention somewhere in the abstract that (a) exon 49 deletion is in-frame and (b) exon 49 skipping would be meant for patients with truncating mutations in exon 49 itself, as this is different than other exon skipping therapeutics.

* Line 41: I would say that dystrophin is severely reduced or absent in DMD, not "totally absent", as many patients do still have a detectable amount of dystrophin.

* Line 96: Were the myoblasts patient-derived or were they engineered to carry the mutation?

* Line 115: It seems the website address is missing.

* Line 121: What was the timing of ASO treatment relative to differentiation? How much medium was used in each well (this would help understand the µL amounts given later in this paragraph)?

* Line 135: How was exon skipping quantified (e.g. densitometry of bands?)

* Line 140: From the phrasing here, I had assumed that these were 5 probands from unrelated families. The supplementary material however indicates that patient D is the daughter of patient E. It would be good to include this in the body of the manuscript, along with any other known family relationships.

* Lines 140-160:

- The most informative patients by far are patients A, C, and E, who are hemizygous or homozygous for the mutation. Patient A is well characterised, but little information is provided about patients C and E. It would be helpful to have at least their ages at examination, presence or absence of weakness, and CK levels.

- Even asymptomatic patients with mutations in DMD can have reduced dystrophin protein levels, probably reflecting a more unstable protein. Can the authors provide a picture of the western blot in patient A and, if possible, a quantification of dystrophin levels in this patient? This would have implications when thinking of the therapeutic potential of the del49 dystrophin isoform and perhaps also the risk of affected patients becoming symptomatic later in life.

- The father of patient E is also an obligate carrier of the mutation (unless there is uniparental disomy or Turner syndrome). Do the authors have any clinical information about him?

* Line 196: I would specify here something like "[...] the pathogenic effect of truncating mutations in exon 49" instead of simply "mutations".

* Lines 219-227:

- Why is the upper band in the treated mutated myotubes so wide compared to the same band in the other experimental conditions? Was this band sequenced? Could it contain a mix of different transcripts arising from different (unintended) splicing outcomes?

- It seems the untreated control is missing in the picture of the WT myotube experiment.

- It would be helpful to provide an additional panel in figure 2 showing the % exon skipping measured for each experimental condition (untreated, AON A, AON B, AON A+B) and cell line.

- Related to the above, a sentence should be added to the text comparing the performance of the different AONs and the efficiency of exon skipping in WT vs mutated myotubes.

- Did the authors try to culture the cells for longer in order to assess dystrophin expression by western blot?

* Line 249: I was very surprised to see that exon 49 deletion had previously been reported to lead to a DMD phenotype. In my experience, some clinicians have a tendency to refer to all patients with neuromuscular symptoms related to DMD mutations as having "DMD", particularly if the patients are still very young. I reviewed the references listed in table S1:

- I was unable to locate the article by Giliberto et al in the list of references or through a pubmed search. Can the authors provide the reference to the paper and confirm that the patient in question has an isolated exon 49 deletion?

- In the other references, I could not find any phenotypic information about the patients that are reported to have "DMD" (e.g. age at loss of ambulation, functional tests, manual muscle testing), nor a definition of what the authors considered to be DMD. Indeed, in some cases, it seems the authors did not have access to phenotypic information that would allow the distinction between BMD and DMD.

- In view of all the above, I think it would be most prudent to add a caveat stating something like "reports of DMD arising from exon 49 deletion did not include any phenotypic information about disease severity".

* Line 255-256: Can the authors expand on the mechanisms they are referring to and perhaps provide some examples from the literature? In most cases, I would expect that the size of the deletion at the genomic level (i.e. the amount of intron 48 and 49 that is deleted) should not affect the resulting mRNA.

* English language: The manuscript would benefit from a review to correct a number of minor mistakes in grammar and English terminology. For example, "gait loss" should be "loss of ambulation", "lozenge" should be "diamond", "primer couples" should be "primer pairs", etc.

Reviewer 2 Report

In this case report like manuscript, the authors described 5 patients with hemizygous, homozygous or Heterozygous exon 49 deletion asymptomatic.  Subsequently they designed anti-sense oligonucleotide to target exon 49 skipping editing.  Overall, the manuscript writing is good. But data are limited. Following are the comments.

(1)    Supplemental figure should be a Figure 2 since you only have 2 figures.

(2)    Despite sequence results showed exon 49 skip in both normal or mutated DMD patients, an IF staining of truncated dystrophin is needed.

(3)    A western blot is necessary to show the shortened dystrophin.

(4)    Why the authors perform exon 49 skip on human normal myotubes?

(5)    Make sure the labels in figures are visible without enlargement of the PDF manuscript.

Reviewer 3 Report

I can agree with the authors that it is the first study which a deletion of the single exon (49) associated with a strikingly normal muscular phenotype was confirmed, but the work is a miss mash.

1. in the introduction, the aim of the study was performed in a very sophisticated way.

2. in introduction, some paragraph does not have the references to support observation/ evidence.

3. materiał and metoda:

a) it is necessary to add the ethic sub-headings

b) material - muscle?, whole blood?

c) please add the catalog number of each reagent, company, city, country

d) 5 patients is unsufficient, unless the authors provide any calculation of the size the group.

e) I cannot see any justification in the paper to use the cell culture.

f) 2.7. sub-section needs to be split to 2 - RNA extraction and RTqPCR.

g) statistical analysis - ????

3. Results were not described in details.

4. Discussion is poor. It needs to add the strengths and limitations of the study; comparison obtaining results with another references etc.

5. number of references are very low.

6. I cannot see the "Ethic statement" and "Written-informed consent".

7. English should be corrected.

Round 2

Reviewer 1 Report

The authors have addressed many of my original concerns, and I think the manuscript is largely suitable for publication, pending correction of a few minor issues:

* In re-reading the methods, it is unclear whether the AON A+B treatment is at the same total dose as AON A and AON B, or the same dose of each AON (i.e. double the total dose).

* I still do not see any quantification  of dystrophin expression in patient A, beyond stating that it is present. The authors should either provide this quantification or acknowledge its absence as a limitation, particularly as they invoke alterations in dystrophin expression as a potential modifier of disease severity in patients with exon 49 deletion.

* I remain very skeptical that introns 48 and 49 could contain regulatory elements of such importance that their loss would result in a DMD phenotype. The well-characterised 45-55 deletion for example involves complete loss of these introns, yet it is nearly always associated with mild BMD.

* I think it would be good to acknowledge that other genetic modifiers of disease severity may playing a role in these patients.

* The authors write that "Furthermore, this could also be explained by low resolution analysis." I think this sentence is difficult to understand for people who are unaware of historical molecular diagnostic techniques at the DMD locus. I would suggest an alternative phrasing, such as "reports of more severe phenotypes associated with exon 49 deletion may also be explained by incorrect ascertainment of the deleted exons due to low resolution analysis."

Reviewer 2 Report

Added figure 2 is great. Strongly suggest bring supplemental table 1 to manuscript, not put as supplemental table since you only have three simple data figures. Table 1 should not have question marker. Maybe use unknown to replace question marker. 

Author Response

We thank you for your help in improving our manuscript.

We leave the decision to the editorial team concerning supplementary table 1

Reviewer 3 Report

In my onion, the paper has not been improved and corrected enough. The authors did not submit the revised paper with "track-changes option". The answers from authors were insufficient. 

Author Response

We regret that this reviewer considers that our manuscript was not improved. We thoroughly replied to all the comments of all reviewers (reviewers 1 and 2 seem now satisfied with this modified version).  

Concerning the track-change file, we apologize for uploading a "saved recently" version.